# Phase-Shift Optimization in AA/PVA Photopolymers by High-Frequency Pulsed Laser

**DOI:** 10.3390/polym12091887

**Published:** 2020-08-21

**Authors:** Daniel Puerto, Sergi Gallego, Jorge Francés, Andrés Márquez, Inmaculada Pascual, Augusto Beléndez

**Affiliations:** 1Departamento de Física, Ingeniería de Sistemas y Teoría de la Señal, Universidad de Alicante, Ap. 99, E03080 Alicante, Spain; sergi.gallego@ua.es (S.G.); jfmonllor@ua.es (J.F.); andres.marquez@ua.es (A.M.); a.belendez@ua.es (A.B.); 2Instituto Universitario de Física Aplicada a las Ciencias y las Tecnologías, Universidad de Alicante, Ap. 99, E03080 Alicante, Spain; pascual@ua.es; 3Departamento de Óptica, Farmacología y Anatomía, Universidad de Alicante, Ap. 99, E-03080 Alicante, Spain

**Keywords:** polymers, optical storage materials, diffractive optics, holography

## Abstract

Photopolymers can be used to fabricate different holographic optical elements, although maximization of the phase-shift in photopolymers has been a challenge for the last few decades. Different material compositions and irradiation conditions have been studied in order to achieve it. One of the main conclusions has been that with continuous laser exposure better results are achieved. However, our results show for the first time that higher phase-shift can be achieved using a pulsed laser. The study has been conducted with crosslinked acrylamide-based photopolymers exposed with a pulsed laser (532 nm). The increment of the phase-shift between the pulsed laser and continuous laser exposure is 17%, achieving a maximum phase-shift of 3π radians and a refractive index shift of 0.0084 at the zero spatial frequency limit, where monomer diffusion does not take place. This allows this photopolymer to be used in large-scale manufacturing.

## 1. Introduction

The use of photopolymers based on polyvinyl alcohol/acrylamide (PVA/AA) to fabricate holographic optical elements [1,2,3,4,5,6] has been a field in constant development in recent decades. Different investigations have studied the recording mechanisms for a wide group of photopolymer compositions and irradiation conditions [7,8,9,10,11,12,13]. Continuous wave (CW) laser processing is well known and widely used [14,15,16], but the experimental requirements (use of lenses, mirrors, spatial filters, etc.) for continuous laser holographic recording systems make their industrial use very expensive.

An alternative technique is holographic copying processes [17] which use a master in direct contact with the photopolymer. This technique is the cheapest to be used in large-scale processes but is incompatible with the use of continuous lasers because both the master and the copy deteriorate. However, with a pulsed laser there is no deterioration phenomenon, and therefore the combination of the copying processes and the pulsed laser can allow holographic recording in industrialized processes.

The good experimental behavior of photopolymer irradiation with a continuous laser has ensured that there are few studies about pulsed laser interactions in photomaterials. For example, the effect of energy, pulse duration and wavelength on different photomaterials has been studied [7,8] as well as different irradiation techniques [9,12,13]. Recently, single pulse holographic recording in PVA/AA has been studied, achieving the record of a hologram of spark discharge under the condition of single pulse exposure [18]. The investigations closer to our study with pulsed lasers have been made in photopolymers based on PVA/AA [19,20,21], although we do not record holograms in this study. Garcia et al.’s studies [19,20] achieve gratings with diffraction efficiencies of 60%, whereas Gallego et al.’s study [21] reaches a diffraction efficiency higher than 85% by adding monomer crosslinker to the material. However, in all three studies the diffraction efficiency and refractive index shift are lower than in continuous laser irradiation for similar layer thicknesses.

In the studies cited above [19,20,21], a pulsed laser of 8 ns pulse duration and up to 10 Hz of repetition rate was used. As described in these studies, the laser irradiation produces free radicals, which react with monomers, initiating the polymer chain growth. When a continuous laser is employed, the next step in the reaction process is the bimolecular combination, where two growing macroradicals come together and finish the process [21]. However, when using a pulsed laser, the initiation of the polymer chain growth is produced during the off time between consecutive pulses [20]. Thus, when the next pulse comes, it creates new free radicals that can react with the polymer chains which are growing and cause an earlier end to the process. This premature process termination causes the formation of a broad molecular-weight distribution that increases the polymer chain length, which is largest as the frequency (or pulse repetition rate) decreases, which reduces the diffraction efficiencies. However, when the laser pulse repetition frequency increases, the free radical rate generation also grows, which enables a larger number of reactions with the monomer molecules. This leads to higher polymerization rates and energetic sensitivity.

Following this argument, we can suppose that at high laser frequencies it is possible to achieve at least the same diffraction efficiencies, polymerization rate, energetic sensitivity and refractive index shift than with a continuous laser. For this reason, this paper will study the refractive index shift in photopolymers based on PVA/AA with an N,N’-methylene-bis-acrylamide (BMA) crosslinker using high repetition rate irradiation (10–500 kHz), achieving for the first time, to our knowledge, a phase-shift 17% bigger than continuous laser irradiation, with a phase-shift of up to a maximum of 3π radians and a refractive index shift of 0.0084.

## 2. Materials and Methods

The phase-shift measurement was made with a real-time interferometric system, which employed a pulsed green laser to modify the material and two CW red laser beams to measure the phase-shift in real time by interference from the two red beams. These two red beams were produced upon illumination of the grating of 5 lines/mm with a He-Ne continuous laser, and the first-order diffracted beams were only used in the real-time measures (Figure 1). Each of the first-order diffraction beams were transmitted through either the irradiated or the nonirradiated areas of the photopolymer, whereas the other orders were blocked. A lens produced the spatial overlap of the two beams after traversing the photopolymer sample. A microscope objective (AN = 0.4 and f = 9.0 mm, Newport M-20X, Irvine, CA, USA) was used to form a magnified image of the interference plane onto the charge-coupled device (CCD) camera (PCO-1600, high dynamic 14-bit cooled camera, Kelheim, Germany). The camera software allowed the exposure time to be controlled, the saturation level to be adjusted and created a delay between consecutive images. Thus, we could register in real time the complete growth of phase-shift by measuring the shift of the interference fringes. In the measurements, an exposure time of 6–8 ms and a delay of 250 ms was used. The refractive index shift was obtained by the equation φ = 2π*Δn/λ, where φ is the phase-shift, Δn the refractive index shift and λ the wavelength of the laser.

The irradiation pulses were obtained from a CW diode-pumped solid-state laser (wavelength of 532 nm and maximum power of 5 Watts, Coherent Verdi V5, Santa Clara, CA, USA) by means of an acousto-optic chopper, which generated a train of pulses with a minimum period of 2 µs (500 kHz) and pulse duration of 1 µs where the period and pulse duration could be controlled independently. The pulse parameters were controlled by a pulse generator (Quantum Composers 9200 series, Bozeman, MT, USA), which sent an electric signal to the acousto-optic modulator (AOM) of the chopper. The beam diameter is Ø = 5.5 mm and the pulse fluence is 55 mJ/cm^2^. The material was exposed for about 240 s until it achieved the maximum phase-shift.

The material composition was selected following the recommendations given by Gallego and Fernandez et al. [22,23]. The photopolymer was composed of AA as polymerizable monomer (AA, 0.84 gr), triethanolamine (TEA, 1.2 mL) as coinitiator and plasticizer, yellowish eosin (YE, 0.7 mL) as dye, PVA as binder, a small proportion of water as additional plasticizer (25 mL of 8% *w/v* PVA) and BMA (0.25 gr) as the crosslinking monomer. The solution was deposited using the force of gravity, distributed with a thickness controller rod of 500 µm on a glass substrate and left for 24 h in the dark to evaporate part of the water, which allowed a layer to generate with enough mechanical resistance to be cut without deforming. This is indicated in reference [23], where the photopolymer composition achieved a 2π radians phase-shift with a thickness layer of 68 µm. In our case, the final layer had a thickness of 95 ± 5 μm, which enabled a higher phase-shift. The refractive index of the pre-exposed photopolymer was 1.477, measured with a refractometer in previous work [24].

In order to avoid the surface diffusion once the irradiation started, which induces an incorrect fitting of the monomer diffusion, the index matching method proposed in references [5,22] was used. This enabled us to decouple the surface and internal changes. To obtain accurate index matching, we have used a commercial baby oil proposed in reference [25], composed of 98% glycerine, with a refractive index of 1.484, very close to the mean of the polymer. This index matching method included a coverplate (microscope slide) to cover the liquid oil and to improve the conservation.

## 3. Results and Discussion

In this study we measured the phase-shift using the experimental setup described above under different irradiation conditions. In this sense, Table 1 shows the polymer phase-shift achieved with different irradiation conditions. It is important to indicate that since a CW laser was used to generate the pulse train, the maxima fluence per second was achieved in continuous mode, whereas the fluence per second decreases as the laser time off increases. Moreover, the fluence per pulse is also affected by this fact, as it decreases when the pulse width decreases. This means that, unlike pulsed lasers, this way of generating pulse trains does not concentrate all the energy in a pulse but maintains the average energy level in the on times and zeroes the energy in the off times.

Although previous studies [19,20] show that, depending on the material composition and the concentration of components, there is an optimum pulse fluence enabling maximum phase-shift with minimum exposure, we have omitted this and have concentrated on exposing the material for enough time until the maximum phase-shift was reached, after which no changes were observed.

As shown by Garcia et al. and Gallego et al. in references [20] and [21], respectively, the maximum refractive index shift is always achieved with CW laser irradiation. However, in our case the maximum phase-shift was achieved in the pulsed irradiation regime (553°, Table 1), being greater than in the CW laser case for almost all conditions of pulsed irradiation. A maximum phase-shift of 3π radians (553°) means that when compared with CW irradiation (2.63π radians or 473°) there is an increment of 17%, and a refractive index shift of 0.0084 is generated, larger than observed in references [20,21]. Therefore, we conclude that, in general, pulsed laser irradiation achieves a greater phase and refractive index shift than CW laser.

Another important point of discussion is about the best fluence per pulse to achieve the maximum phase-shift or refractive index shift. In this case, we did not observe any relation between fluence per pulse and phase-shift. For example, at fluences close to 0.5 µJ/cm^2^ (first column, Table 1), different values of maximum phase-shift were measured, from 494° to 533°. Similarly, no relation between fluence per second (second column, Table 1) and phase-shift was detected. For example, the maximum phase-shift values range from 458° to 533° at fluences close to 35 mJ/cm^2^. For this reason, we are inclined to think that the greatest influence on phase-shift is given by the period and the width of the pulse. Moreover, we want to emphasize that the fluences per pulse used in this article are lower (µJ/cm^2^) than those used in references [19,20,21] (mJ/cm^2^), which means that it is possible to start the photopolymerization process at high frequencies (kHz) with lower fluences per pulse than when working at low frequencies (Hz).

In order to analyze the influence of pulse-time parameters on phase-shift, we have made Figure 2 and Figure 3, which collect all the information included in Table 1. Figure 2 shows phase-shift as a function of pulse period and pulse width, whereas Figure 3 shows phase-shift as a function of pulse period and pulse-off time.

Observing both figures, the first fact to highlight is that just one pulsed irradiation achieves a phase-shift lower (458°) than CW irradiation (473°). This is produced with a period of 100 µs, a pulse width of 70 µs and a pulse-off time of 30 µs, showing that even for high pulse rates (10 kHz) lower phase-shifts than those of CW irradiation can be produced. Comparing this result with those obtained in the references [20,21], with 10 Hz frequency and 8 ns of pulse duration, we can conclude that the connection between both laser irradiation conditions is the long pulse-off time. Therefore, when the pulse-off time is long, at least more than 30 µs, the phase-shift (polymerization rate) is lower than in CW irradiation because the free-radical concentration decreases to near zero between consecutive pulses. This causes the radical-monomer reaction to stop, so the growing polymer chain reacts with new free radicals generated by the next pulses and causes an earlier end to the process.

The second point of interest of the figures is the three irradiation conditions with a phase-shift slightly higher than CW irradiation (473°): a period of 20 µs, a pulse width of 10 µs and a pulse-off time of 10 µs (495°); a period of 12 µs, a pulse width of 10 µs and a pulse-off time of 2 µs (500°); and a period of 10 µs, a pulse width of 9 µs and a pulse-off time of 1 µs (494°). For phase-shifts of 500° and 494°, the ratios between pulse width and pulse period are higher than 80%, leading to a behavior similar to CW irradiation because the free-radical rate remains almost continuous between consecutive pulses. However, the explanation of the similar behavior to CW irradiation during a 495° phase-shift can be supported by a long pulse-off time (10 µs).

The other groups of irradiation with a medium and high phase-shift are as follows: (*medium*) a period of 7 µs, a pulse width of 5 µs and a pulse-off time of 2 µs (515°); a period of 4 µs, a pulse width of 2 µs and a pulse-off time of 2 µs (522°); a period of 2 µs, a pulse width of 1 µs and a pulse-off time of 1 µs (517°); (*high*) a period of 15 µs, a pulse width of 10 µs and a pulse-off time of 5 µs (533°); a period of 10 µs, a pulse width of 5 µs and a pulse-off time of 5 µs (530°); and a period of 10 µs, a pulse width of 3 µs and a pulse-off time of 7 µs (553°). The common points between them are a pulse-off time lower than 10 µs (≤7 µs), and ratios between pulse width and pulse period lower than 80% (70–30%). The main difference is that the pulse-off time is between 5 µs and 7 µs for a high increment of phase-shift, and for a medium increment of phase-shift the pulse-off time is between 1 µs and 2 µs.

Following the explanation about photopolymerization processes given first by Shelkovnikov et al. [13] and later by Garcia et al. and Gallego et al. [20,21], it is complicated to explain both medium and high increments in the phase-shift for pulsed irradiation regarding the CW irradiation. However, the previous paragraphs give us some key points about this. As Decker indicates [26], a saturation effect, in which a partial recombination of the excited photoinitiator is rendered again in the starting dye, is produced once the concentration of the excited photoinitiator reaches a certain high value. Continuous exposure with the CW laser does not favor the partial recombination of the excited photoinitiator because there is no pulse-off time that stimulates this recombination, which reduces regeneration of the starting dye and therefore results in less total free radicals. With a pulsed laser, if the pulse-off time is higher than 1 µs (our maxima time resolution) and the ratio between pulse width and pulse period is lower than 80%, then a recombination of the excited photoinitiator is favored and, therefore, there is a high regeneration of the starting dye. This allows a higher total free-radical formation and thus bigger polymerization rates. However, for a long pulse-off time (≥10 µs), although the amount of regenerated starting dye also increases, the pulse-off time is long enough to allow the radical-monomer reaction and the polymer chain growth, which can react with the free radicals generated by the next pulses and cause an early end to the process.

Therefore, we can conclude that the minimum irradiation conditions to achieve a higher phase-shift than CW irradiation are a ratio between pulse width and pulse period lower than 80% to allow a high regeneration of the starting dye, and a pulse-off time between 1 µs and 10 µs to allow a chain-reaction process and the polymer radical growing, while avoiding an earlier end of process. This means that laser frequencies higher than 20 kHz (period lower of 50 µs) are necessary to achieve phase-shifts higher than CW irradiation, as it is not possible to keep these previous conditions at lower frequencies. Moreover, the irradiation conditions required to achieve the maximum phase-shift are a pulse-off time of between 5 µs and 7 µs and ratios between pulse width and pulse period lower than 80%.

## 4. Conclusions

The effects of irradiation parameters in the phase-shift in photopolymers based on PVA/AA with monomer crosslinker have been studied in this paper. In particular, the effect of high-frequency irradiation compared with low-frequency of previous studies has been checked. The main conclusion is that with high frequencies that are more than 20 kHz (period of 50 µs), higher phase-shift values (up to 3π) than CW irradiation can be achieved with increments of up to 17% and a refractive index shift of up to 0.0084. This is because the pulse-off time is long enough to allow a high regeneration of the starting dye but not too long to avoid the reaction between growing polymer chain and new free radical, leading to higher polymerization rates. When the ratio between pulse width and pulse period is higher than 80%, the phase-shift has similar behavior to CW irradiation. Moreover, when the pulse-off time is higher than 30 µs, the phase-shift is lower than CW irradiation. This better behavior of the PVA/AA-based photopolymer with pulsed irradiation will allow its use in holographic copying processes, which improves the potential utilization of this photopolymer in large-scale industrialized processes.

## Figures and Tables

**Figure 1 polymers-12-01887-f001:**
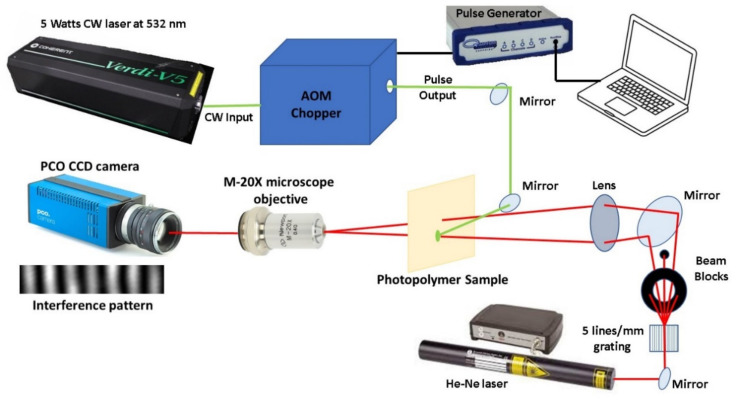
Experimental setup of the irradiation system.

**Figure 2 polymers-12-01887-f002:**
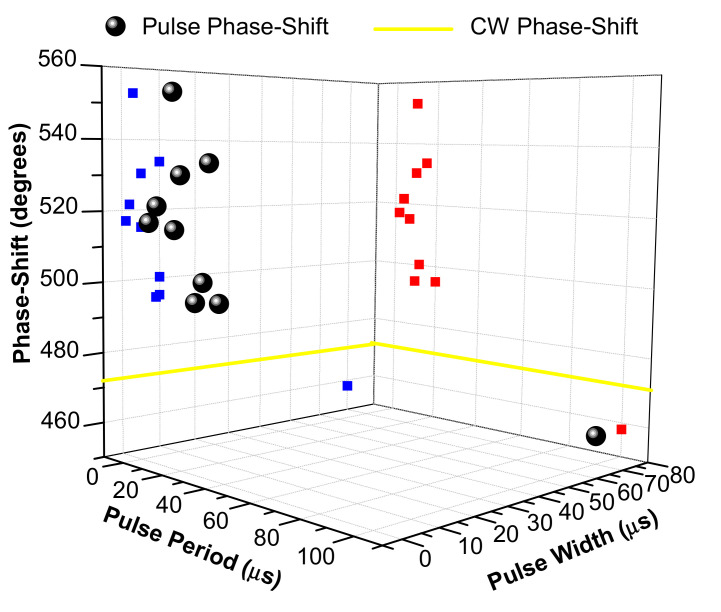
Polymer phase-shift after pulsed irradiation as a function of pulse period and pulse width (●), projection of the phase-shift in the pulse period axis (█) and pulse-off time axis (█). Polymer phase-shift after continuous irradiation (━).

**Figure 3 polymers-12-01887-f003:**
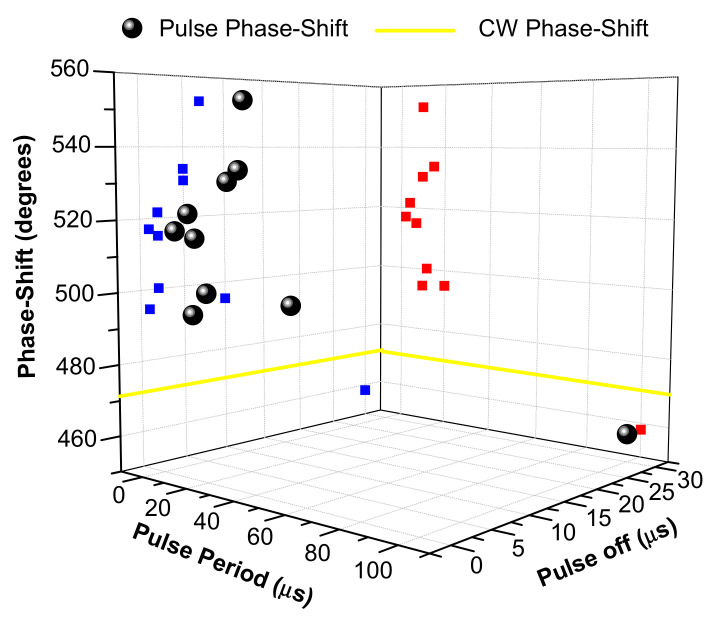
Polymer phase-shift after pulsed irradiation as a function of pulse period and pulse-off time (●), projection of the phase-shift in the pulse period axis (█) and pulse-off time axis (█). Polymer phase-shift after continuous irradiation (━).

**Table 1 polymers-12-01887-t001:** Polymer phase-shift after continuous wave (CW) irradiation and after pulsed irradiation as a function of pulse period, pulse width and pulse fluence.

Fluence Per Pulse (J/cm^2^)	Fluence Per Second (mJ/cm^2^)	Pulse Period (µs)	Pulse Width (µs)	Phase-Shift (Degrees)
55 × 10^−3^	55	CW	CW	473
3.48 × 10^−6^	35	100	70	458
0.52 × 10^−6^	26	20	10	495
0.51 × 10^−6^	34	15	10	533
0.52 × 10^−6^	43	12	10	500
0.48 × 10^−6^	48	10	9	494
0.27 × 10^−6^	27	10	5	530
0.26 × 10^−6^	37	7	5	515
0.15 × 10^−6^	15	10	3	553
0.10 × 10^−6^	25	4	2	522
0.05 × 10^−6^	25	2	1	517

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
