# Peer review of "Phase-Shift Optimization in AA/PVA Photopolymers by High-Frequency Pulsed Laser"

_polymers, 2020, doi:10.3390/polym12091887_

Round 1

Reviewer 1 Report

The authors inscribe holograms in photopolymers using pulsed lasers. The study claims that the material used, i.e. AA/PVA could be used in large scale manufacturing. These two-directions claims are not sufficiently supported in the discussions since each of the two directions needs to be put in a its proper context. While I dont't see why pulsed lasers are better than continuous waves lasers given the absorption in the same visible band, and no nonlinear effect is discussed, the materials context referred by the authors mentions silver halide only. Other materials, e.g. azo-polymers, can be used for holographic recording using both CW and pulsed laser, in single steps and require no development phases (OSA Continuum 1, 668–681 (2018). Appl. Opt. 55, 259-268 (2016)). The authors should put their work in proper context, and should take into considerations the above points to improve their manuscript.

The paper ca not be recommended for publication in its present form.

Author Response

Point-by-point reply to the reviewer’s comments

Points raised by referee 1

The authors inscribe holograms in photopolymers using pulsed lasers.

From the referees' indications, we have detected that they have some doubts regarding the aim and experimental method of paper, perhaps proven by an insufficient explanation of it.

The aim of the article are described in the lines 17-18 of the abstract “However, our results show that higher phase modulation can be achieved using pulsed laser” and  64-68 of the introduction “For this reason, in this paper we study the refractive index modulation in photopolymers based on PVA/AA…..”. Therefore, in this article holographic elements are not produced and we also do not use an experimental system to record holographic elements. We measure the phase shift induced in the material after irradiation with a pulsed laser and for this we use a beam of a continuous laser (He-Ne) to cross a non-irradiated area and another He-Ne beam to cross the irradiated area, and after crossing the material the beams interfere to measure the phase shift with the change of the interference fringes.

To emphasize this, we have included the next text  “…although we don't record holograms in this studio”. Furthermore, we explain the experimental setup in the lines 71-88 of the article but to clarify and avoid whatever doubts we have included the next text "The phase-shift measurement has been made with a real-time interferometric system, which employs pulsed green laser to modify the material and two CW red laser beams to measure phase shift in real time by interference from the two red beams.

Furthermore, we think that some of the referees' doubts may be due to the use of the words "phase modulation" and “phase depth” in this type of experiment, which is commonly used in the hologram recording. For that reason, we have changed, even in the title, these words by "phase-shift" that is more appropriate. In general, phase shift is related to phase modulation when sinusoidal pattern is used as recording process in holography, but in the experiments presented in this work phase-shift is more appropriate.

The study claims that the material used, i.e. AA/PVA could be used in large scale manufacturing. These two-directions claims are not sufficiently supported in the discussions since each of the two directions needs to be put in a its proper context.

The aim of the article is to compare the phase-shift achieved after the irradiation with CW and pulsed laser. One of the best hologram recording techniques for manufacturing large-scale holographic optics is holographic copying processes, but this technique must use a pulsed laser (see lines 34-38) to avoid the setup elements deterioration.

Since we have demonstrated that the phase-shift achieved after irradiation with pulsed laser is greater than that achieved with continuous laser, we can conclude that holographic copying processes with pulsed laser can be used to record holographic optical elements at large-scale with better properties than with techniques that use CW laser.

While I dont't see why pulsed lasers are better than continuous waves lasers given the absorption in the same visible band, and no nonlinear effect is discussed,

We base the best behaviour of the pulsed laser compared to the CW laser on the optimization the use of the dye to start the photopolymerization process, which allows achieving a greater phase-shift (see lines 190-205). This is due to the saturation effect described by Decker in [21], in which a partially recombination of the excited photoinitiator regenerate the starting dye. Our point is that for pulse off time between 1 µs and 10 µs a recombination of the excited photoinitiator is favoured and, therefore, a higher regeneration of the starting dye than CW irradiations.

Furthermore, since the pulsed laser is generated by a CW laser (see lines 81-88), the maximum laser energy is achieved with the CW laser mode and therefore the possibility of non-linear effects is the same or more in CW than in pulsed irradiations.

the materials context referred by the authors mentions silver halide only. Other materials, e.g. azo-polymers, can be used for holographic recording using both CW and pulsed laser, in single steps and require no development phases (OSA Continuum 1, 668–681 (2018). Appl. Opt. 55, 259-268 (2016)).

Following the referee recommendations we have included the reference [10] “Sekkatz Z., Vectorial motion of matter induced by light fueled molecular machines, OSA Continuum 2018, 1-2, 668-681”

The authors should put their work in proper context, and should take into considerations the above points to improve their manuscript.

As can see in the answers above we have taken into considerations the recommendations of the referee.

Author Response

Point-by-point reply to the reviewer’s comments

Points raised by referee 3

It should be pointed out that 9 out of 21 references are self-citations. I would recommend significantly expand the list of references by including references on the research carried out by other researchers as well as pay more attention to the published date and make preferences for the recent publications (last 5 years).

Following the referee recommendations we have included new references regarding the use of PVA/AA in holographic optical elements recording (references [3] and [4]). However, the employ of pulse laser in holographic optical elements recording isn´t so usual, for that reason we use self-citations, including the same citations of used by Zhao et all. in the reference https://www.sciencedirect.com/science/article/abs/pii/S0167577X14016395

I would also suggest include the reference

https://www.sciencedirect.com/science/article/abs/pii/S0167577X14016395

Following the referee recommendations we have included “Recently, it has been studied single pulse holographic recording in PVA/AA, achieving the record of a hologram of spark discharge under the condition of single pulse exposure [16]”

and put discussion how results achieved in the present research advances the state-of-the-art.

Following the recommendations of the referees, we have included new references ([3], [4], [10], [16]) and some comments to discuss how the results obtained in the present investigation advance the state of the art. In this sense, we have included in the abstract this red words “However, our results show for the first time that higher phase-shift can be achieved using pulsed laser”, and in the introduction the next sentence “achieving for the first time, to our knowledge, a phase-shift 17% bigger than continuous laser irradiations, with a phase-shift of up to a maximum of 3π radians and a refractive index shift of 0.0084

Line 1. Authors say “Photopolymers can be used to fabricate different diffractive and holographic elements”.

I would propose to change the terminology and use “holographic optical elements” instead of “diffractive and holographic elements” as mentioned diffractive element fabrication in photopolymers is based on holographic techniques.

Following the referee recommendations we have changed “diffractive and holographic elements” to “holographic optical elements”

Line 28. Authors say “The use of photopolymers based on polyvinyl alcohol/acrylamide (PVA/AA) to fabricate recording elements such as holographic [[1],[2]] and diffractive devices [[3],[4]] ”…

Unfortunately, I can’t find any sense in this wording. I would suggest rephrase the sentence to have scientific sense.

Also, Authors put references of the research on holographic optical elements developed in PVA/AA-photopolymer by holographic techniques as two separate type of devices a) holographic, b) diffractive.

Following the suggestion for Line 1, I would suggest combine references 1-4 and define them as “holographic optical elements”. Also, I would suggest including few more recent references as currently cited references are at least 9 years old.

Following the referee recommendations and such as we have indicated in the first and second comments, we have changed “diffractive and holographic elements” to “holographic optical elements” and included new references ([3] and [4]).

Line 49. I would recommend following corrections. “As described in them, the laser photopolymerization irradiation is producesd by the free radicals generation, which reacts with the monomers initiating the polymer chain growth (radical growing).”

We have attended the referee recommendations.

Line 72. I would recommend following corrections “… a He-Ne continuous laser to generate the two beams that create holographic pattern with a grating spatial frequency of 5 lines/mm”.

From the referees' indications, we have detected that they have some doubts regarding the aim and experimental method of paper, perhaps proven by an insufficient explanation of it.

The aim of the article are described in the lines 17-18 of the abstract “However, our results show that higher phase modulation can be achieved using pulsed laser” and  64-68 of the introduction “For this reason, in this paper we study the refractive index modulation in photopolymers based on PVA/AA…..”. Therefore, in this article holographic elements are not produced and we also do not use an experimental system to record holographic elements. We measure the phase shift induced in the material after irradiation with a pulsed laser and for this we use a beam of a continuous laser (He-Ne) to cross a non-irradiated area and another He-Ne beam to cross the irradiated area, and after crossing the material the beams interfere to measure the phase shift with the change of the interference fringes.

To emphasize this, we have included the next text  “…although we don't record holograms in this studio”. Furthermore, we explain the experimental setup in the lines 71-88 of the article but to clarify and avoid whatever doubts we have included the next text "The phase-shift measurement has been made with a real-time interferometric system, which employs pulsed green laser to modify the material and two CW red laser beams to measure phase shift in real time by interference from the two red beams.

Furthermore, we think that some of the referees' doubts may be due to the use of the words "phase modulation" and “phase depth” in this type of experiment, which is commonly used in the hologram recording. For that reason, we have changed, even in the title, these words by "phase-shift" that is more appropriate. In general, phase shift is related to phase modulation when sinusoidal pattern is used as recording process in holography, but in the experiments presented in this work phase-shift is more appropriate.

Reviewer 3 Report

1.Could Authors give more information about used AA/PVA  photopolymers?

2.How to obtain the phase depth of 3π radians and the refractive  index modulation of 0.0084 ?

3. How to  know the diffusion effect doesn't take place in this experiment?

4. The importance of this experiment should be added in the introduction?

5.How to obtain the Phase depth in this experiment? The measurement of  Phase depth should be clearly described in the method.

Author Response

Point-by-point reply to the reviewer’s comments

Points raised by referee 2

  1. Could authors give more information about used AA/PVA photopolymers?

AA/PVA photopolymers are widely used in holography. The material is referenced with the contributions of some research groups, see ref. [2, 4, 5, 15]. This photopolymer has been studied with different dyes, and crosslinkers. In our paper the chemical composition is similar to the presented in ref. [21] and it was used to store diffractive optical elements such as optical vortexes using spatial light modulator as a master in Fernández R, Gallego S, Márquez A, Neipp C, Calzado EM, Francés J, Morales-Vidal M, Beléndez A. Complex Diffractive Optical Elements Stored in Photopolymers. Polymers. 2019; 11(12):1920. doi:10.3390/polym11121920

  1. How to obtain the phase depth of 3π radians and the refractive index modulation of 0.0084?

We described in the “Materials and methods” section with “The phase-shift measurement has been made with a real-time interferometric system” that we measure the phase shift induced in the material after irradiation with a pulsed green laser using a beam of a continuous laser (He-Ne) to cross a non-irradiated area and another He-Ne beam to cross the irradiated area, and after crossing the material the beams interfere to measure the phase shift the movement or shift of the interference fringes. To emphasize in the description, we have added the following sentence in the text “Thus, we can register in real-time the complete growth of phase shift by measuring the shift of the interference fringes

Furthermore, to clarify and avoid whatever doubts about the experimental setup (lines 71-88) we have included the next text "The phase-shift measurement has been made with a real-time interferometric system, which employs pulsed green laser to modify the material and two CW red laser beams to measure phase shift by interference from the two red beams.

Regarding the “refractive index shift”, we agree to referee, the measurement mode is not described in the article. Therefore, we have added the next sentence "The refractive index shift is obtained by the equation f=2π*Dn/l, where f is the phase-shift, Dn the refractive index shift and l the wavelength of the laser.”

Moreover, we think that some of the referees' doubts may be due to the use of the words "phase modulation" and “phase depth” in this type of experiment, which is commonly used in the hologram recording. For that reason, we have changed, even in the title, these words by "phase-shift" that is more appropriate.

  1. How to know the diffusion effect doesn't take place in this experiment?

 The diffusion effects doesn´t affect our experiments in this paper, because we use a uniform green laser irradiation with wide area, 5.5 mm of the spot size, and the phase changes are measured at the centre with one of the red beams. Diffusion takes place when there is a spatial gradient in the chemical compound concentration, that exists only in the boundaries of the green spot. These boundaries are too far, taking into account the monomer diffusion values for AA/PVA materials 10-11 cm2/s and the exposure times, the monomer travels less than 1 mm. Furthermore in the “Materials and methods” section (lines 104-109) we discuss the surface diffusion (apparent diffusion), which is avoided using an index matching method proposed in Ref [5] and [21].

  1. The importance of this experiment should be added in the introduction?

Following the referee recommendations we have included in the abstract this red words “However, our results show for the first time that higher phase-shift can be achieved using pulsed laser”, and in the introduction the next sentence “achieving for the first time, to our knowledge, a phase-shift 17% bigger than continuous laser irradiations, with a phase-shift of up to a maximum of 3π radians and a refractive index shift of 0.0084

  1. How to obtain the Phase depth in this experiment? The measurement of Phase depth should be clearly described in the method.

We have answered this question in the second question of the referee.

Round 2

Reviewer 1 Report

Further to my context comment, the authors included only one reference, and I have suggested 2. This suggestion is meant to improve the author's work context. The authors should also include the Appl. Optics Reference.

my previous comment:

the materials context referred by the authors mentions silver halide only. Other materials, e.g. azo-polymers, can be used for holographic recording using both CW and pulsed laser, in single steps and require no development phases (OSA Continuum 1, 668–681 (2018). Appl. Opt. 55, 259-268 (2016)).

Author Response

Following the referee recommendations, we have included the reference [11] “Sekkatz Z., Optical tweezing by photomigration, Appl. Opt. 2016, 55-2, 259–268”

We want to thank the referee for his constructive participation in the discussion of this article, which has helped to improve it.

Reviewer 2 Report

Authors have provided the improved version of the manuscript that has more comprehensive sections such “Introduction” and “Materials” and allows the reader to understand the novelty of the research.

I would suggest the manuscript for the publication after introducing the minor changes as suggested below.

Line 28. I would suggest removing “recording elements” in the sentence

 The use of photopolymers based on polyvinyl alcohol/acrylamide (PVA/AA) to fabricate recording elements such as holographic optical elements [[1],[2]..

Line 51. I would suggest using refractive index shift instead of index shift.

Line 79.  A He-Ne continuous laser is used to generate the two beams with a grating of 5 lines/mm (Fig. 1).

Very confusing sentence. I would suggest to rephrase.

These two red beams are produced upon illumination of the grating of 5 lines/mm with a He-Ne continuous laser and the first-order diffracted beams are only used in the real-time measurements (Fig. 1).

Line 133

I would suggest the following correction

… and the concentration of components

Line 160

A typo. “At” must be instead of “a”

Figure 2 and 3. To follow the terminology in the paper, phase shift must be used instead of phase depth.

Author Response

Point-by-point reply to the reviewer’s comments

Points raised by referee 3

Authors have provided the improved version of the manuscript that has more comprehensive sections such “Introduction” and “Materials” and allows the reader to understand the novelty of the research. I would suggest the manuscript for the publication after introducing the minor changes as suggested below.

We want to thank the referee for his constructive participation in the discussion of this article, which has helped to improve it.

Line 28. I would suggest removing “recording elements” in the sentence

The use of photopolymers based on polyvinyl alcohol/acrylamide (PVA/AA) to fabricate recording elements such as holographic optical elements [[1],[2]..

Following the referee recommendations, we have eliminated “recording elements such as” in the previous sentence.

Line 51. I would suggest using refractive index shift instead of index shift.

Following the referee's recommendations, we have included "refractive" on line 51 and have revised the article to make sure "refractive index" always appears.

Line 79.  A He-Ne continuous laser is used to generate the two beams with a grating of 5 lines/mm (Fig. 1).

Very confusing sentence. I would suggest to rephrase.

These two red beams are produced upon illumination of the grating of 5 lines/mm with a He-Ne continuous laser and the first-order diffracted beams are only used in the real-time measurements (Fig. 1).

Following the referee's recommendations, we have included the above proposed change.

Line 133 I would suggest the following correction

… and the concentration of components

Following the referee's recommendations, we have included the above proposed change.

Line 160 A typo. “At” must be instead of “a”

Following the referee's recommendations, we have included the above proposed change.

Figure 2 and 3. To follow the terminology in the paper, phase shift must be used instead of phase depth.

Following the referee's recommendations, we have included the two new figures with the correct axis labels “phase-shift”.

Reviewer 3 Report

The author have replied the comments, and it can be accepted now.

Author Response

We want to thank the referee for his constructive participation in the discussion of this article, which has helped to improve it.
